# The Critical Importance of Spatial and Temporal Scales in Designing and Interpreting Immune Cell Migration Assays

**DOI:** 10.3390/cells10123439

**Published:** 2021-12-07

**Authors:** Jennifer Frattolin, Daniel J. Watson, Willy V. Bonneuil, Matthew J. Russell, Francesca Fasanella Masci, Mikaila Bandara, Bindi S. Brook, Robert J. B. Nibbs, James E. Moore

**Affiliations:** 1Department of Bioengineering, Imperial College London, London SW7 2AZ, UK; j.frattolin@imperial.ac.uk (J.F.); d.watson@imperial.ac.uk (D.J.W.); willy.bonneuil15@imperial.ac.uk (W.V.B.); 2Centre for Mathematical Medicine and Biology, School of Mathematical Sciences, University of Nottingham, Nottingham NG7 2RD, UK; matthew.russell@nottingham.ac.uk (M.J.R.); bindi.brook@nottingham.ac.uk (B.S.B.); 3Institute of Infection, Immunity and Inflammation, College of Medicine, Veterinary Medicine and Life Sciences, University of Glasgow, Glasgow G12 8TA, UK; Francesca.FasanellaMasci@glasgow.ac.uk (F.F.M.); Mikaila.JayaweeraBandara@glasgow.ac.uk (M.B.); robert.nibbs@glasgow.ac.uk (R.J.B.N.)

**Keywords:** cell migration, chemotaxis, chemokine, leukocytes, scale, live cell tracking

## Abstract

Intravital microscopy and other direct-imaging techniques have allowed for a characterisation of leukocyte migration that has revolutionised the field of immunology, resulting in an unprecedented understanding of the mechanisms of immune response and adaptive immunity. However, there is an assumption within the field that modern imaging techniques permit imaging parameters where the resulting cell track accurately captures a cell’s motion. This notion is almost entirely untested, and the relationship between what could be observed at a given scale and the underlying cell behaviour is undefined. Insufficient spatial and temporal resolutions within migration assays can result in misrepresentation of important physiologic processes or cause subtle changes in critical cell behaviour to be missed. In this review, we contextualise how scale can affect the perceived migratory behaviour of cells, summarise the limited approaches to mitigate this effect, and establish the need for a widely implemented framework to account for scale and correct observations of cell motion. We then extend the concept of scale to new approaches that seek to bridge the current “black box” between single-cell behaviour and systemic response.

## 1. Introduction

Advances in experimental techniques have led to significant improvements in our understanding of the processes that govern adaptive immunity. These processes rely on directed cell migration to facilitate precisely timed encounters between multiple cell types. Cells are directed by endogenous signals, such as chemokines, as well as environmental cues like the structure of the extracellular matrix (ECM) and interstitial flow. The migration of immune cells from the periphery to lymph nodes (LNs) and other secondary lymphoid organs (SLOs), as well as within the SLOs, play a critical, intrinsic role in adaptive immunity. In homeostasis, a small number of semi-mature dendritic cells (DCs) constantly sample tissues and migrate to lymphatic vessels to be transported to LNs. Here, they join a network of resident antigen-presenting cells (APCs) that sample tissue-derived lymph and encounter high densities of lymphocytes that entered the LN primarily from the blood. Inflammation, tissue damage, or infection induces DCs to rapidly mature and causes them to migrate *en masse* to lymphatic vessels. Whether semi-mature or mature, DCs process the proteins they acquired in the tissue and present peptide antigens on major histocompatibility-complex (MHC) proteins to T cells in the LNs. The LN microarchitecture guides DC localisation and the migratory tracks of lymphocytes, increasing the probability of T cells finding their cognate antigen [1,2]. Semi-mature DCs migrating from healthy tissue present antigens derived from harmless or self-proteins and induce T cell anergy or deletion to eliminate potentially autoreactive T cells and thereby maintain immune tolerance. However, mature DCs, which may carry antigens from pathogen-derived proteins, can activate cognate naïve T cells, resulting in their clonal expansion and their differentiation into effector cells armed to destroy the pathogen.

Naïve T cells enhance the probability of encountering cognate antigen by migrating within LNs and other SLOs and by moving between SLOs via the lymphatic and blood-vessel networks [3]. After a residency period of 6–24 h in SLOs, naïve T cells egress via efferent lymph or blood vessels, leaving space for other T cells to enter. Activated T cells remain in SLOs for longer to give them time to complete their maturation before they leave as effector cells and home to inflamed tissue. Naïve T cells primarily enter LNs through high endothelial venules (HEV) [4,5,6], but they can also enter via afferent lymph when LNs are arranged in series. Likewise, memory T cells and regulatory T cells, depending on their precise phenotype, can enter LNs via HEVs or from tissue-draining afferent lymph [7].

Chemokines and their receptors are the essential messengers of leukocyte trafficking [8,9]. To guide their short-term migration, leukocytes polarise their morphology and orient their movement according to differences in chemokine concentration across their diameter (1–10 μm). They must integrate these with adhesion cues, antigens, and the local ECM topology to follow longer-range chemotactic signals [6,10]. This type of migration is commonly observed in interstitial DC migration in peripheral tissue and LNs. While the chemotactic response of cells to chemokines is only relevant to some lymphatic trafficking, chemokines can also influence cell behaviour in other ways. For example, with naïve T cells, CCR7 ligands on HEVs stimulate T cell arrest on the HEV, while in the T cell zone of the paracortex, these chemokines enhance speed but do not regulate cell directionality, a process known as chemokinesis [5].

Our understanding of adaptive immune response has been aided by advancements in the experimental techniques used to study cell migration. As our understanding of adaptive immunity continues to expand, the temporal and spatial scales over which these phenomena occur has gotten larger, but the temporal and spatial scales of the assays used to investigate these effects have not kept pace. There is an assumption in the field that, with modern imaging techniques, leukocytes move sufficiently slowly such that imaging can occur frequently enough to accurately capture the cell path. However, this assumption is almost entirely untested; as a result, the influence of scale on the characterisation of cell migration is overlooked and under-appreciated in the literature. Experimental limitations of live cell imaging constrain quantitative measurements in ways that can obscure the true nature of migratory behaviours. This can place a limit on image acquisition frequency, which, if insufficient, can result in an underestimation of cell velocity and an overestimation of chemotaxis. Furthermore, small fields of view increase the probability that cells enter or leave during the acquisition time or before another image can be taken. Insufficient spatial and temporal resolutions within migration assays can significantly influence quantitative statements about cell migration or cause subtle cell behaviours to be missed.

In recent years, multiple works, and reviews of those works, have focused on the inference of the mechanisms of adaptive immunity, but there has been limited review of the methods employed to study cell migration in the adaptive immune response. In this review, we do not discuss the molecular mechanisms involved in leukocyte migration, as other reviews have made substantial efforts to summarise and detail this [11,12,13,14]. Here, we focus on the different methods of studying cell migration and contextualising the impact of scale-based phenomena on estimations of leukocyte migration. In this review, we firstly summarise the various experimental techniques implemented to quantify cell motility and the different approaches of measuring immune cell migration with respect to scale. Then, we review how scale can affect cell migration parameters and confound the interpretation of cell tracking results. Finally, we discuss how some of these technical constraints can be addressed and how we can bridge the scale gap between single-cell and organ-scale behaviour. A hybrid approach of experimental assays and mathematical models allow for the characterisation of immune cell migration under a variety of conditions that would be significantly more difficult or unfeasible in experimental assays alone [15,16].

## 2. Measuring Migration: Techniques and Motivation

Observation and quantification of leukocyte migration can be achieved through a variety of in vivo, ex vivo, and in vitro techniques. In general, in vivo assays represent a gold standard in understanding complex leukocyte migration. In the past, this was limited to endpoint assays [17,18,19,20], where the location of cells is not tracked during the assay. Recently, advances in microscopy have resulted in the development of in vivo techniques, such as intravital imaging with two-photon microscopy [21,22,23], which do allow for direct analysis of single-cell dynamics and interactions with the surrounding environment, generally referred to as a “kinetic assay”; this type of assay can also be extended to in vitro techniques. Here, a summary of the current state of the art with regards to migration assays is presented. An overview is provided of how leukocyte migration is measured in both endpoint and kinetic assays at varying scales, their associated advantages and limitations, as well as a review of common motility parameters used to describe cell migration.

### 2.1. From Endpoint Assays to Single-Cell Analysis: The Evolution of Migration Assays

#### 2.1.1. Endpoint Assays

##### Boyden Assay

The Boyden, or transwell, assay was one of the first techniques developed to assess the chemotactic behaviour of leukocytes [24]. Cells are incubated inside a transwell with a chemoattractant in the well underneath, resulting in cell migration through the pores of the transwell chamber and into the well (Figure 1). Cells are then analysed by microscopy of the lower surface of the transwell, counting with a haemocytometer, or by flow cytometry [17,25,26,27]. Several mathematical models were developed to estimate the motility coefficient and chemotaxis of cells during this assay, as well as assay modifications to improve these measurements [28,29,30]. The advantages of the Boyden assay include its relative simplicity and the wide variety of pore sizes commercially available to analyse different cell types [31]. The assay can be conducted with the cells in suspension or on top of a matrix, with both wild-type and gene-knockout cells. However, the Boyden assay is unable to maintain a stable chemokine gradient long-term or to fully recreate the biological conditions found in vivo during migration. There is also the potential for functional or phenotypic alterations in ex vivo and cultured cells, which can impact the migratory behaviour of cells. Lastly, the results of a Boyden assay are population-averaged and, as a result, cannot capture cellular heterogeneity. 

##### Adoptive Cell Transfer Assays

Adoptive cell transfer assays require the injection of fluorescently labelled donor cells into recipient mice. These assays allow the investigation of large-scale, macroscopic migration of leukocytes from the site of injection to an end location of interest, such as a LN or SLO. They can be differentiated by the injection location, which can be subcutaneous (footpad or tail base), intravascular/intravenous, or intralymphatic (Figure 2). After subcutaneous injection of labelled donor cells, the draining LNs are analysed, typically by flow cytometry to determine the number of transferred cells [18,25,32,33,34,35,36,37,38,39,40,41,42,43,44]. Microscopic examination of fixed LN sections after immunohistochemical or immunofluorescence processing is also commonly used to assess donor cell presence and location [32,44,45,46,47,48]. Adoptive cell transfer via intravenous injection can be used to assess homing from the bloodstream to other tissues [49]. However, intravenous injection leads to systemic and circulatory cell migration, targeting multiple organ systems, which requires the analysis of multiple organs by flow cytometry to determine the number of labelled cells they contain [19,20,40,48,50,51,52,53]. Intralymphatic injection is a technique that transfers donor cells directly into the afferent lymphatic vessels of a LN [20,54] and is used to monitor traffic of bone marrow-derived DCs (BMDCs) and other cells in and out of the subcapsular sinus and through the parenchyma. One of the primary advantages of this technique is the removal of cell migration from the skin into the lymphatics, reducing the complexity of the migratory process to just entry from the lymph into the LN. After a fixed period of time, the LN is fixed, stained, and imaged.

The primary advantage of these assays is the ability to manipulate donor cells and recipient mice, through fluorescent labelling, genetic manipulation, and the use of congenic mice, enabling the molecular mechanisms of cell migration to be studied in a large-scale macroscopic assay. However, there is the potential for an altered cell phenotype resulting from the extraction, culture, and labelling of the donor cells, which can influence cell migratory behaviour [13]. Furthermore, both subcutaneous and intravascular injections require a non-physiological number of donor cells. In contrast, intralymphatic injections require fewer cells to be transferred, but it is a technically challenging procedure, requiring anaesthesia that may influence immune cell migration and lymphatic pumping [55]. Cannulation of the afferent lymphatic vessel may also result in an inflammatory response by the recipient mouse.

##### Dermal Sensitisation Assay

Dermal sensitisation is a well-established migration assay used to evaluate the trafficking of cutaneous DCs to LNs in response to a sensitising agent. In its simplest form, the abdomen, back, or, ears of mice are inflamed (Figure 2), and the number of DCs in the draining LN are quantified by flow cytometry. Fluorescein isothiocyanate (FITC) can be included in the sensitising agent and the trafficked DCs identified by the presence of the stain [32,34,35,36,41,48,56,57,58]. This assay is advantageous as it is technically straightforward, extensive DC migration can be induced [59], and knockout mice can be used to analyse the effect of gene deletion on cell migration. There are several limitations to this technique, however. In particular, it depends on the successful uptake of FITC to track cells from skin to LN. More recently, photoswitchable mice have been used to mitigate this limitation [60]. Furthermore, this technique does not allow for homeostatic cell migration to be investigated.

##### Split-Ear Assay

The split-ear migration assay is an ex vivo technique used to track the attraction of BMDCs towards dermal lymphatics (crawl-in assay) or the egress of endogenous DCs out of the dermis (crawl-out assay). The ears of mice are split into dorsal and ventral halves and then incubated with BMDCs for a crawl-in assay or with media only for a crawl-out assay. For a crawl-in assay, the ear halves are fixed, stained, and then imaged with confocal microscopy [37,40,46,61,62,63,64,65]. Images are analysed to determine the percentage of labelled DCs within lymphatic vessels or the dermis, as well as the distance the DCs are from a lymphatic vessel. For a crawl-out assay, the media are harvested and analysed by flow cytometry to determine the number of DCs that migrated out of the ear dermis [17,18,23,40,56,57,58]. However, the use of ex vivo tissue has its limitations. The protocol for preparing the explanted ear half may result in a change in its cellular structure [66]. In addition, the removal of the tissue from its in vivo environment will limit its ability to reproduce all relevant biological conditions, which may result in altered DC behaviour [13].

#### 2.1.2. Kinetic Assays

The primary disadvantage of all endpoint assays is that it is not possible to directly visualise the cells of interest during their migration. As a result, there is no information available to probe the mechanisms determining the different migratory processes along that pathway or to assess cellular heterogeneity within a population. However, with the development of advanced microscopy techniques, such as two-photon microscopy, and the creation of novel in vitro models, such as microfluidic chips, direct single-cell tracking is now attainable with in vivo, ex vivo, and in vitro kinetic assays. While the advent of single-cell tracking and analysis has led to considerable advances within the field of immunology, it is limited by the restricted spatial resolution of available microscopy techniques. This limits the ability to extrapolate the observed cell behaviour to larger scales, including the organ-scale. Here, we review the primary in vivo, ex vivo, and in vitro kinetic assays within the literature and summarise their specific advantages and disadvantages.

##### Intravital Microscopy

Intravital microscopy (IVM) is a well-established in vivo technique to analyse single-cell behaviour during leukocyte trafficking and immune response within the native in vivo environment [66]. Genetic manipulation of both the donor cells and recipient mice can be implemented, including knockout, reporter, and transgenic mice. IVM can be used in a variety of anatomical locations in murine models [21,23,32,37,67,68]; cells of interest are labelled and injected into recipient mice, including subcutaneously, intravenously, intralymphatically, and intraperitoneally. Specific to intralymphatic injection, the synchronous arrival of cells into the subcapsular sinus is advantageous to track cell location over time, which is not possible with a footpad injection. Some time after adoptive transfer, which can vary from 1 h to several days, mice are anaesthetised and prepared for intravital imaging, with the tissue of interest surgically exposed, most commonly a LN or immobilised ear pinnae (Figure 3). Imaging is typically conducted with a two-photon or inverted confocal microscope. The resulting cell tracks are determined either with a commercial software, such as Imaris (Oxford Instruments, Switzerland) or Volocity (Quorum Technologies Inc., Canada) or with an in-house custom method. Parameters of interest are estimated from the tracks, including the mean or instantaneous velocity (μm/min), the displacement (μm), the chemotactic index, the turning angle, the persistence, and the motility coefficient (μm^2^/min).

While IVM has led to significant advancements in the field of immunology, it has its limitations. IVM is a technically challenging procedure that requires a specialised imaging setup. Only a small region of cell migration can be imaged at any point in time, typically on the order of hundreds of microns, and, depending on the microscope used, there can be a restricted *z*-depth due to working-distance limitations. In addition, this technique requires prolonged anaesthesia of the mouse, which is known to impact immune cell dynamics [55]. Furthermore, surgery to expose LNs or other areas of interest can cause considerable inflammation, which can influence cell behaviour. Like other in vivo assays, it is difficult to finely control experimental conditions in IVM to elucidate preferred cell behaviour, such as controlled chemotactic cell motion.

##### Ex Vivo Migration Assays

Ex vivo kinetic assays use high-resolution microscopy to observe the migration of immune cells in explanted tissue, such as LNs, spleen, or ear halves. When utilising explanted LNs, the tissue can be imaged intact or in slices. Prior to excision, labelled cells of interest can be adoptively transferred through intravenous, intralymphatic, or subcutaneous injection. LNs are explanted and prepared for imaging after a set amount of time following cell transfer, which can vary from 40 min to 24 h [54,69,70,71]. For the split-ear assay, the infiltrating DCs within the dermal lymphatics of the ear are imaged microscopically [37,43,65].

There are distinct advantages to such ex vivo migration assays. They allow the direct visualisation of cell dynamics in the native-tissue environment, without the use of anaesthesia or invasive surgery. They also allow the genetic modification of the donor cells and the recipient mouse, as well as chemical manipulation at the time of transfer, such as immunisation. However, the implications of the ex vivo environment, such as the absence of fluid flow (blood, lymph, or interstitial) or lack of physiologic solute concentrations and transport on cellular function are not well understood [13,66]. In addition, the methods of tissue preparation can potentially have a significant impact on the resulting cellular behaviour.

##### Migration Assays on 2D Coated Surfaces

Migration assays on 2D coated surfaces are a commonly implemented in vitro technique to directly observe cell motion [31]. The complexity of such assays can vary significantly from cell migration on coated plates to more complex microfluidic chips. The coated-plate technique utilises a flat surface coated with an extracellular matrix component or a cell adhesion molecule, such as ICAM-1-Fc [31], plus a coating of chemokine. The Dunn chamber is a chemotaxis chamber consisting of multiple concentric rings (Figure 4a) [72]. Cells are cultured on a coated coverslip that is inverted on top of the chamber, after which the outer and inner rings are filled with chemokine and media, respectively. This allows for the creation of a chemokine gradient across the 20 μm gap between the coverslip and the bridge separating the source and buffer rings. The under-agarose migration assay utilises a cast agarose mould with two punched holes; one is the responder hole, filled with cells, and the other is the attractor hole, filled with chemokine (Figure 4b).

Microfluidic chips are a commonly used technique for migration assays with a wide range of designs (Figure 4c) [73,74,75,76]. The most common approach utilises a polydimethylsiloxane (PDMS) base with the desired channel geometry and is cast on a wafer etched by photolithography. The PDMS is then bonded to a coverslip. For 2D migration assays, the cell channel of the chip is typically coated with fibronectin or ICAM-1-Fc, with the adjacent channels connected to a fluid source that provides chemokine solution or cell-culture media. In all 2D migration assays described, cells are visualised over time as they migrate towards the chemokine source. It should be noted that under-agarose and Dunn-chamber assays can also be implemented as endpoint assays without the use of live-cell tracking.

These migration assays have some advantages. Overall, they minimise the need for complex animal procedures. They can be used with a wide variety of cell types, including cells isolated from genetically modified mice [13]. In addition, in vitro migration assays allow the isolation of chemokine-directed cell migration under controllable conditions, facilitating the study of non-directed cell motion, which is not easily isolated in vivo. However, there are limitations. Chemokine gradients are not easily produced in migration assays on coated plates [31], and while the under-agarose assay and the Dunn-chamber assay facilitate the development of chemokine gradients by diffusion, they do not allow for long-term gradient stability (Figure 4d). As a result, microfluidic chips have a distinct advantage over other in vitro migration assays. The resulting gradients have long-term stability and can be shaped by the chip geometry and modified during experiments (Figure 4d). In addition, microfluidic chips can be used to distinguish between chemotaxis and haptotaxis in vitro [74]. However, the chemical and physical limitations of the in vitro testing environment are the most significant. An in vitro assay cannot fully reproduce the biological conditions found in vivo. Furthermore, the 2D testing environment implemented in these assays is a simplistic representation of 3D tissue microenvironments [77].

##### Migration Assays in 3D Matrices

In vitro migration assays using 3D matrices are designed to represent the migratory motion of cells through sections of ECM. They are typically implemented with hydrogels based on collagen, which is a major component of ECM [78]. There are two primary methods by which hydrogels can be incorporated into in vitro migration assays. One approach uses cell-encapsulated hydrogels polymerised in a well plate, dish, or custom holder, with chemokines typically placed on top of the hydrogel to form a soluble gradient within the matrix [20,37,40,46,62,63]. A modification of that approach uses hydrogels mixed with chemokine, after which the cells are placed on top of the hydrogel and allowed to migrate through the matrix towards the chemoattractant [34]. The second technique for 3D migration assays uses microfluidic chips, where the hydrogel is injected into a channel with specifically designed posts to entrap the hydrogel (Figure 3). There are other chip designs, but this is one of the most common. Channels adjacent to the hydrogel region are connected to a fluid source, from which chemokines or media are continuously supplied to facilitate chemokine gradient formation. In both techniques, after hydrogel polymerisation, a microscope is used to image cell migration during the assay [31]. Cell images are then processed either with commercial or in-house segmentation software, and the desired motility parameters, such as the cell velocity or the chemotactic index, can be estimated.

The use of a 3D matrix-based migration assay has some distinct advantages over a 2D system. Cells exhibit different migratory behaviour in 2D versus 3D environments, and it has been shown that there is a significant dependency of cell migration on the structure of ECM [79]. Hydrogels can be tailored to have mechanical properties similar to those of native ECM. In addition, many hydrogels can be customised and functionalised with other ECM components known to influence cell migration, such as proteoglycans and other matrix proteins [77]. Glycosaminoglycans can bind to chemokines, creating gradients of bound chemokine within the matrix [80,81]. In microfluidic chips, this facilitates both chemotactic (from soluble gradients) and haptotactic (from bound gradients) motion of leukocytes. This allows the elucidation of cell migratory behaviour in tightly controlled environmental conditions. However, there are limitations with these techniques. 3D migration assays conducted in a well plate or dish do not allow stable or consistent chemokine gradient formation, which will affect chemotactic cell motion. In addition, the limited working distance of the microscope objective may allow only a small section of hydrogel to be imaged, depending on the height of the hydrogel and the thickness of the bottom of its container. As with 2D in vitro migration assays, the primary limitation of these techniques is the limited representation of the in vivo environment, although the use of a functionalised hydrogel marks a significant improvement in comparison to its 2D assay counterpart [77].

### 2.2. How Can Cell Migration Be Quantified In Vivo and In Vitro?

Leukocyte migration can be estimated at a variety of scales from macroscopic to microscopic. The techniques available to evaluate cell migration are highly scale-dependent, where the type of assay conducted dictates how leukocyte migration can be assessed. Endpoint assays allow for the observation of large-scale cell migration at a macroscopic level, where a significant number of cells move from a known origin to an end location. The spatial scale from one observation point to another limits the techniques available to quantify the resulting cell migration as the cell path is not tracked during the assay. As a result, it is not possible to elucidate cell behaviour in real-time or accurately estimate migration parameters such as cell velocity, displacement, or chemotactic index. Instead, the number of cells of interest identified at the end location are reported or presented as a migration index (the number of cells present divided by the total number of cells transferred). In contrast, kinetic assays allow for microscopic observation of single cells in real-time, with the ability to track individual cells with high-resolution microscopy, albeit within a small field of view. The resulting cell tracks can be used to quantify cell motility parameters, such as the cell displacement (μm), the velocity (μm/min), and the chemotactic index, as well as to characterise the type of cell migration observed.

Characterising cell behaviours in kinetic assays requires appropriate recognition of the uncertainties in quantifying migratory patterns, as illustrated in Figure 5, where a cell has been tracked at an imaging interval of Δt. The cell has moved from its origin to another position some distance away after t=nΔt, where *n* is the number of imaging intervals. Cell-motility parameters of interest can be quantified based on the resulting cell tracks. However, every quantitative statement made about the motion of cells is dependent on an assumed model for that motion. The resulting understanding of the motion of cells depends on the assumptions taken as to what occurred during the unobserved aspects of the assay and which aspects of the motion can be characterised as constant; the higher the fidelity of these models of motion, the more representative their measured parameters and the stronger their applicability in interpreting or predicting cell behaviours. In this example, a cell migrates continuously in time during the assay, as depicted by the blue path (Figure 5). In kinetic assays, the position of the cell is not known at all times; rather, it is only known as a series of discrete positions at specific times, indicated by the apparent path in red. The cell path is approximated by assuming that the cell moved in a straight line between two consecutive observations and that it moved constantly in the time between them. Therefore, the frequency at which a cell’s position is known (the frame rate) will influence the fidelity of the apparent path.

A variety of cell-motility parameters can be estimated from the resulting tracks, including cell displacement and mean or instantaneous cell velocity. The persistence of a cell, defined as the tendency for a cell to continue in its current direction, can be assessed by determining a persistence index (also commonly referred to as a meandering index, a straightness index, or a confinement ratio). This is defined as the total cell displacement divided by the apparent-path length. It is distinct from the chemotactic index, which is conceptually defined as the ratio of chemotactic motion to total cell motion. Most works represent chemotactic motion as the cell displacement in the direction of a known chemokine gradient (defined in Figure 5 as $\hat{d}$) and the total motion as either the total displacement or the apparent-path length. However, there is not a consensus within the field for the definition of the chemotactic index, and other definitions of the chemotactic index do exist. A forward migration index has been used to describe the chemotactic motility of cells [34], with a similar definition to that described previously. At least one study has presented the chemotactic index as the dot product of the cell direction and the direction of interest by calculating the cosine of the angle between them [62]. Chemotactic indices are commonly quantified from in vitro cell tracks, where known chemokine gradients can be established and maintained. In vivo assays require a direction of interest to be assumed in order for a chemotactic index to be estimated; as a result, persistence indices are reported more often.

Another technique commonly used to assess leukocyte migration are plots of mean-squared displacement (MSD) versus time [82]. They can be used to infer the type of migration exhibited by the ensemble behaviour of many cells. For a sufficiently large number of particles undergoing Brownian motion, there will be a linear relationship between the MSD and time. The slope of this line is the motility coefficient, in μm^2^/min, which is a metric used to describe the tendency of a cell to migrate from its origin in a random fashion. This is analogous to the diffusion coefficient for Brownian particles. The motility coefficient can be determined using the following formulae, $M=d^{2}/4t$ for 2D motion, where *d* is the displacement of each cell at time *t* from its initial position [83], or $M=d^{2}/6t$ for 3D motion [84]. Deviation from the slope indicates non-Brownian behaviour, such as chemotactic motion. This technique has appeared with some of the earliest in vivo observations of leukocyte motion, and its use continues to characterise cell behaviour in modern works [85,86,87,88].

## 3. Relating Cell Motion to Spatial and Temporal Observation Scales

### 3.1. Phenomena of Scale in Kinetic Assays

Cell migration is a complex phenomenon, where cells may pause or change directions due to external constraints or internal biological processes. MSD plots represent the preferred tool to characterise the behaviour of cell migration (Figure 6). A qualitative inspection of these plots can reveal deviations from Brownian motion; definitions of such migratory behaviours are provided in Figure 7. Super-diffusive behaviour is visible as an upward inflection in a MSD plot, where the scaling exponent is greater than one (Figure 6a). In contrast, sub-diffusive behaviour is characterised by a downward inflection in a MSD plot, where the scaling exponent is less than one. MSD plots are the most commonly used method to assess cell migratory behaviour, but they rely purely on the identification of an inflection, and its interpretation is not always straightforward. In reality, there are a number of factors at different scales that can affect the character of an MSD curve, which may obscure the true nature of cell migration. Here, we give an overview of sub-diffusive and super-diffusive motion and explore how different factors can influence these behaviours. We also contextualise how the spatial and temporal scales can influence quantitative statements on cell migration and demonstrate the resulting power-law scaling of cell motion in experimental assays.

#### 3.1.1. The Causes and Appearance of Sub-Diffusive Motion

While sub-diffusive behaviour could theoretically have a number of causes, the preponderance of literature on the matter has considered two main sources of origin; first, the physical effects that can constrain motion creating sub-diffusive behaviour at certain scales and, second, the observation artefacts that can create the impression of sub-diffusive behaviour if the scale is not considered carefully. A fundamental assumption in the interpretation of MSD plots is that all cells have moved unhindered and can be observed no matter how far they move from their origin. In reality, cells are in a finite container and are observed by an instrument with a limited field of view. The length scale of the container and the time scale over which a cell will remain within the observation window will constrain assay design and affect analysis.

In intravital and ex vivo microscopy, physical structures that can constrain cell motion are often geometrically complex, can be difficult to see, and can potentially span multiple scales. This can make their impact on the observed MSD plots difficult to predict. The motion of cells may be constrained physiologically within a vessel or organ, in which case the effect of the container on the motion must be considered. Constrained motion can also be observed when a cell must pass around other cells during its migration. For example, Hugues et al. observed that T cells exhibited constrained migration when in proximity to DCs along their path trajectory [89]. Constrained motion may also partially conceal other cell behaviours, such as directed motion. Vroomans et al. demonstrated that the chemotactic behaviour of T cells to DCs can be muted by spatial exclusion; i.e., for every T cell that moved towards a DC, another is forced to leave to make room for it, resulting in a decrease in the overall directional bias of the T cells [90]. When it comes to in vitro observations, homogeneity of the migration media can be reproducibly achieved, and the dimensions of any container are known a priori. However, the displacement of the cell will never exceed the length scale of the container, which creates the appearance of sub-diffusive behaviour. Therefore, microfluidic devices in particular need to be large enough to allow the cells to exhibit the behaviours relevant to the phenomena being studied.

For most published assays, the volume or area available for cells to travel is larger than that observed. This can create the impression of sub-diffusive behaviour that would not be present with a larger observation window or a shorter assay time. If cells leave the observation window during an assay, they are no longer tracked. As time progresses, this biases the cells included in the MSD plot to be only those whose displacement is less than the length scale of the microscope, creating the impression of sub-diffusive behaviour. During in vivo tracking of T cells during parasitic encephalitis, Harris et al. state that they limited their reported observations to 10 min to prevent the over-representation of less-motile cells [86]. Theoretical studies on particles diffusing in cages have suggested a point of inflection marks the scale at which the constraint dominates the motion, and this approach has been used experimentally by Fricke et al. to mitigate this phenomena [87,91]. Friedl et al. reported the number of cells that left the observation window, though this practice is not widespread [92].

#### 3.1.2. The Causes and Appearance of Super-Diffusive Motion

The most commonly considered cause of super-diffusive motion is directed motion, the simplest of which is ballistic, where cells move in a straight line toward their attractant at a constant velocity. In this model, the displacement increases linearly with time, which results in a quadratic curve on an MSD plot and super-diffusive motion (Figure 6c). Pure chemotaxis is an example of ballistic motion. In practice, cells continue to migrate randomly but are biased over time. This drift-diffusion motion of cells is also super-diffusive, though this is less apparent on an MSD plot. Cells that are undergoing both chemotaxis and random migration will have the appearance of drift-diffusion motion. A common technique used to confirm that the observed deflection of the MSD plot is due to chemotaxis is to look for a directional bias in the cell tracks. Here, the observation scale is critical as directed motion will only be evident if the observation scale is smaller than the gradients that cause directed motion.

Super-diffusive behaviour is not solely associated with directed motion. Super-diffusive flights, such as Lévy flights, are also a common cause of super-diffusive behaviour, which are found at all time scales. In super-diffusive flights, the distance a cell moves before changing direction, called a jump length, is drawn from a heavy-tailed distribution. This is in contrast to Brownian motion, where jump lengths are drawn from a Gaussian distribution resulting in a linear relationship between MSD and time. The difference between these two behaviours on a MSD plot is shown in Figure 6b. Unlike in directed motion, the resulting random walks of super-diffusive flights have no directional correlation, i.e., they are as likely to go in any direction as any other, but they produce super-diffusive motion. This kind of behaviour produces efficient searches and has been proposed as another solution to the “needle in a haystack” problem of T cell-APC contact presented by Cahalan et al. [93]. This super-diffusive behaviour requires careful analysis to separate it from directed motion. Harris et al. reported a Lévy flight behaviour in T cells during encephalitis toxoplasmosis [86]. While MSD plots were indicative of Brownian motion, the log-log MSD plots showed straight lines with a gradient indicative of super-diffusive motion. The distribution of jump lengths more closely agreed with a Lévy distribution than a Gaussian distribution. Identifying any of these behaviours does not preclude other behaviours. To rule out additional types of directed motion, Harris et al. verified that the aspect ratios of cell tracks produced by such Lévy flights agreed with those found. Fricke et al. expanded on this approach with a more rigorous series of statistical methods that suggested that, in the LN, the motion is better described with a series of steps drawn from a log-normal distribution producing a super-diffusive flight that was not a Lévy flight [87].

Another cause of super-diffusive behaviour is persistence, which is similar to directed motion. At its smallest scale, the fundamental mechanisms of cell motion can cause a persistence. In migrating cells, filopodia extend from the lamellipodium in the initial stage of motion. This creates an arm-like projection filled with cytoplasm, called a pseudopod, which drags the cell to a new location. These structures are then reorganised to form a new lamellipodium, and the process continues in a new direction [94]. At a sufficiently fine time scale, a strong dependence between the current and future direction will be observed (Figure 8a). On an MSD plot, this phenomenon can be observed as a transition from ballistic to Brownian motion at a certain time scale called the persistence time, $t_{p}$, and a length scale called the persistence length, $l_{p}$, as shown in Figure 6c. Other examples of persistence exist at other scales, including cell migration facilitated by contact guiding. Cells migrating through ECM will preferentially move along sub-structures within the ECM, in particular, collagen fibres, creating a persistence at a scale equal to the size of the fibres [88].

Insufficient temporal resolution can result in the appearance of super-diffusive motion, particularly for multi-modal cell motion. Multi-modal motion can produce a variety of observed velocities with different temporal correlations; this can complicate apparent paths that are reconstructed from a series of discrete positions. For example, in vivo observations of T cells in the LN have shown that T cells have periods of dormancy before and after moving a certain distance in a constant direction [85]. Most models assume that cells travel at a constant velocity and in a straight line between two subsequent positions; this fallacy is revealed when the observation interval includes multiple behaviours. This causes multi-modal velocities to appear as a series of different distributions depending on the observation scale. An illustrative example is shown in Figure 6d–f.

The history of the “needle-in-the-haystack” question of how T cells find antigen-presenting DCs within lymph nodes is an excellent example of the importance of careful interpretation of cell tracks. Speculation of how T cells find antigen-presenting DCs predates the first reported in vivo observations of T cell motion, and the resulting data suggested T cell chemotaxis would facilitate the search for antigen-presenting DCs. The presence of CCL19 and CCL21 within the lymphoid organs, the up-regulation of CCR7 after DC maturation, and reduced DC migration to T cell zones in CCR7 deficient mice suggested that they use this signal to migrate to T cell regions to increase DC–T cell interaction. In addition, mature DC expression of the naïve T cell attractant CCL18 suggested chemotaxis may encourage initial antigen presentation [95]. The up-regulation of several activated T cell attractants, such as CCL19 and CCL22, during DC maturation suggested they may encourage additional DC contact for the latter stages of activation and proliferation [96]. However, when the first two-photon in vivo studies allowed cell motion to be observed directly, the upward inflection characteristic of directed motion was absent from the MSD plots [21,83]. This led to Cahalan et al. suggesting a new paradigm that an undirected stochastic mechanism drives T cell–APC contact [93], rather than chemotaxis. However, a review conducted by Bajénoff et al. challenged this notion, stating that there was no conclusive study that demonstrated the percentage of cells required to be chemotactic for a deflection in the MSD plot to be observed [97]. This was further emphasised by Castro et al. [98], who stated that subtle biases in cell tracks could cause directed cell motion to appear random and concluded that careful analysis of mixed-effects models is required to tease out these subtle changes in biased motion.

The ability to differentiate between these variations in super-diffusive behaviours is reliant upon subtle changes in MSD plots. The observation scale, or temporal resolution, that is used will have a significant impact on the ability to identify such changes. Furthermore, quantitative estimations of cell motion will also be affected by the temporal resolution. Most flights, including all Brownian and Lévy flights, will result in a power law scaling for apparent-path length with a temporal scale. That is, as the time between observations decreases, the total distance a cell is perceived to have travelled increases (Figure 8b) and more closely follows the actual cell path. As a result, this will also cause power-law scaling for the cell velocity and the chemotactic index (Figure 8c), where cell velocity is underestimated, and the chemotactic index is overestimated, as Δt increases. When the observation scale is less than the persistence time, this phenomenon disappears, and the velocity and the chemotactic index become independent of the observation scale (Figure 8c). This scale will be heavily dependent on the individual experiment and is not widely reported.

#### 3.1.3. Variation of Temporal Scale in Experimental Assays

There is considerable variability in the temporal scales used for kinetic assays. This inconsistency in temporal scales can have a significant influence on estimated migration parameters. The fractal nature of a cell’s apparent path above the persistence scale will induce a power-law scaling of any length-based parameters derived from it. In particular, there is a power-law scaling of cell velocity and chemotactic and persistence indices at different observation time intervals. As the imaging interval increases, cell velocity is underestimated, and chemotactic and persistence indices are overestimated (Figure 8c). In some instances, the temporal scale selected is technically constrained, for instance, as a result of microscopy limitations. A compromise is often required between the desired image volume, the slice thickness, and the time needed to obtain the image stack. This effect is less prominent in 2D where the required imaging time is significantly reduced. Here, we highlight the variability in temporal scales reported within the literature for different in vivo, ex vivo, and in vitro kinetic assays within the LN and surrounding periphery and demonstrate the effect of these differing imaging intervals on the estimated parameters of interest, including the cell velocity and the chemotactic index.

The imaging interval, or the temporal scale, implemented in in vivo and ex vivo assays varies depending on the cell type and the imaging location (Table 1). For intravital imaging of LNs, an imaging interval between 15–60 s is typically utilised. More specifically, for DCs, a temporal scale of 15–30 s is selected [23,37,41,42,43,67,99,100,101], though as high as 50 s has been reported [32]. In contrast, an imaging interval of 20–60 s is implemented for T cells [22,52,68,102,103,104,105,106,107,108], though observations have been conducted at scales as low as 10 s [21]. Lastly, for B cells, a temporal scale of 15–30 s has been used in previous studies [20,51,53,103,107,109,110,111], with as low as 0.5 s reported [112]. In ex vivo assays, an imaging interval between 10–30 s is commonly implemented within literature for imaging of an excised LN or spleen [54,69,70,71,83,85,87,89,113,114]. In split-ear assays, DCs are typically imaged at an interval between 15–30 s [37,65], though scales as high as 150 s have been implemented [43].

In contrast, the imaging intervals selected for 2D and 3D in vitro assays vary between 10–120 s (Table 1). In 2D assays, such as under-agarose and Dunn-chamber assays, a temporal scale of 15–60 s is typically used for DCs [23,37,46,64,75,76,101,115,116,117,118], though imaging intervals as high as 180 s [119], 300 s [34], and 600 s [45,120] have been reported. In contrast, for 3D assays, including those conducted in microfluidic chips, the typical temporal scale varies between 10–120 s for DCs [32,37,40,43,46,47,61,62,63,121,122,123,124,125,126,127], with intervals as low as 5 s [128]. Intervals above 120 s and as high as 300 s are also common for DC tracking [26,34,129,130,131,132]. In general, T cells and B cells are studied less routinely in in vitro assays than DCs, and, as a result, only limited information on imaging intervals for these cells exists. Temporal scales between 10–60 s have been reported for T cells [52,71,116,117,118,126] and between 3–120 s for B cells [20,37,115].

To illustrate the effect of the imaging interval on the migration parameters of interest, we produced similar log-log plots to that of Figure 8c, utilising experimental data of the cell velocity and the chemotactic index extracted from the references reported in Table 1. To maintain a relevant comparison, data extraction was restricted to wild-type mice, and, for in vitro assays, velocity measurements were restricted to chemotactic cell motion. We further isolated by assay type (in vivo/ex vivo and in vitro), as well as cell type. Despite the inherent variability of the extracted data amongst the different assay techniques, it is evident that a correlation exists between the cell velocity and the imaging interval, regardless of the assay or the cell type (Figure 9). As predicted, a general trend was observed where the cell velocity was underestimated with increasing Δt. For instance, in a study conducted by Clatworthy et al. [32], DCs were tracked every 50 s with intravital imaging of the popliteal LN. The corresponding DC velocity was approximately 1.8 μm/min. In contrast, in another study of intravital imaging of the popliteal LN conducted by Mempel et al. [99], the corresponding DC velocity was 6.6 μm/min, with DCs tracked every 15 s. Similar variations in velocities were observed with different temporal scales for all cell types. When assessing the effect of the temporal scale amongst in vitro chemotaxis assays, a similar behaviour is observed with the DC velocity and the chemotactic index to that depicted in Figure 8c. Chemotaxis is overestimated with increasing Δt, while cell velocity is underestimated (Figure 10). For example, for DCs encapsulated in 3D matrix exposed to CCL19, a chemotactic index of approximately 0.2 was determined when cells were tracked every 60 s by Brown et al. [62], whereas a chemotactic index of 0.81 was estimated by Lammermann et al. [37] with an imaging interval of 120 s.

### 3.2. Phenomena of Scale in Endpoint Assays

While in this section we have focused on the effects of scale on live-cell tracking data, temporal and spatial resolutions can also influence the results of endpoint assays. Unlike in live-cell tracking, endpoint assays involve measuring the cumulative motion of a large number of cells over a longer period. While this limits the amount of information that can be discerned and only allows for statistics at the population scale, it removes a large number of the opportunities for the observation scale to have a distorting effect. However, the spatial scale still needs to be considered with relation to any studied phenomena, and temporal effects can still appear as a result of assay design. A good example of this is found in in vitro migration assays with unsteady chemotactic gradients, such as the Boyden, under-agarose, and Dunn-chamber assays (Figure 4). In these assays, the gradient is not constant, and so the time period over which an assay is conducted changes the mean gradient to which the cells are exposed. The scale of this effect is closely related to the diffusivity of the chemoattractant; a more motile chemoattractant will reach a uniform gradient more quickly than a less diffusive chemoattractant. This restricts the ability to compare assay results using chemoattractants with very different diffusivities as even two assays of the same dose and time may not be fairly compared.

There exists a wide variety of length scales within endpoint assays, from in vivo studies covering migration distances of centimetres to the millimetre distances in Dunn-chamber and under-agarose assays, all the way down to Boyden assays with a filter thickness of tens of microns. In all of these assays, the diameter of the cells themselves remains the same, and the cell migration observed over great distances is likely to be affected by phenomena inconsequential over a few cell diameters.

## 4. Discussion

### 4.1. Contextualising the Effects of Scales: Establishing a Scale-Cognisant Framework

In 2005, Halin et al. [66] summarised the recent developments of in vivo microscopy of lymphocyte trafficking and concluded that common procedures were needed across research teams to better understand the impact of different measurement parameters in three-dimensional intravital imaging. Since then, researchers have exploited the power of two-photon microscopy and pushed both intravital and other assay techniques to their physical limits in an attempt to better understand the mechanisms governing immune cell migration. While procedures have expanded in both magnitude and complexity over the past decade, a consensus has yet to be reached regarding the critical measurement parameters of 3D live cell tracking. Substantial variability of measurement parameters remains within the literature, with a lack of consistency in the temporal and spatial scales selected for in vivo, ex vivo, and in vitro alike.

This is problematic for multiple reasons. It suggests a lack of appreciation of the deleterious effects of under-sampling. Insufficient temporal and spatial resolutions during cell tracking can have significant implications on the characterisation of leukocyte migration, which can result in misrepresentation of important physiologic processes and behaviours, such as an overestimation of the chemotactic index, the obfuscation of directed motion, or misconstruing cell-arresting behaviour. This can have significant implications when elucidating the underlying biology of immune cell behaviour or the effect of genetic knockouts. Such studies often rely on small changes in leukocyte migration to establish causal links, such as contextualising behaviour between different lymphocyte subsets [68] or understanding the role of adhesive integrins on lymphocyte migration [71]. Correlation between the temporal and spatial scales and the measured parameters limits comparisons between studies and increases the complexity of a potential meta-analysis of the field. It is therefore critical that widespread adoption of studies cognisant in the effects of scale is pursued. However, only a few examples of scale-cognisant studies currently exist within the literature [86,87]. Harris et al. is an excellent example of a scale-cognisant study [86], which limited cell tracking to only 10 min to avoid biasing their data set to less-motile cells and which grouped data by imaging intervals to avoid making comparisons between different temporal scales.

The question remains of how best to design and implement a scale-cognisant study. There are some general guidelines that can be followed when conducting live cell tracking. These include reducing the imaging interval to increase the accuracy and fidelity of the cell path, as well as limiting the total observation time and maximizing the field of view. A set of pilot studies can be conducted varying sampling and spatial parameters to project scaling behaviour on migration metrics of interest; these studies could assess the persistence time to ensure it is not significantly less than the imaging interval. However, it is important to realise that the required spatial and temporal resolutions are highly cell-specific and assay-dependent. Cells migrate in different ways and speeds depending on their local environment, and these effects are more prominent with some cell types than others. Furthermore, potential cellular heterogeneity within a population must also be considered, and it is critical to image at the speed of the fastest cell within the population of interest.

However, while these are good guiding principles, they will not always be practical or feasible experimentally. Technological constraints linked to microscopy will often limit the ability to fully implement such an approach. Even if technological advances could overcome instrument limitations, there remain the issues of photobleaching and phototoxicity. While these issues are well known in visible light confocal microscopy, the preponderance of the field uses two-photon microscopy, and these issues do not seem as significant when the incident light is infrared. However, quantitative studies of the effects of phototoxicity on these cell types are limited and not well documented. Remarks on this subject in the literature likely represent a small fraction of the work done as most preliminary studies involve optimising for laser power and dwell time. However, authors rarely describe the steps they took to mitigate or allow for these effects, and, as a result, their true impact remains unknown. Label-free tracking presents an opportunity to avoid these potentially deleterious effects but is not always experimentally feasible.

As a result of these limitations, it may not be possible to completely mitigate the effects of scale during experimental design, and, consequently, consideration of the effects of scale should be included in the data interpretation and analysis of studies. Care should be taken when defining exclusion criteria for cell tracks; there is a wide variety of exclusion criteria within the literature, such as a minimum duration of cell movement and a critical cell velocity or track length [23,37,104,108,134]. Setting cell-velocity and track-duration limits too high can bias data to slower, less-motile cells. In addition, due to the power-law scaling of cell velocity, exclusion criteria cannot be universal but must depend on the specific imaging parameters. Power-scaling plots, such as that shown in Figure 10, can be implemented to better understand the effect of scale on estimated parameters within assays, while presenting data and conclusions within the context of this limitation. When comparing studies, in particular, velocity magnitudes or the chemotactic index, the effects of scale must be considered, and relationships need to be established across experimental results using differing scales.

These practical limitations could be overcome through the use of mathematical modelling. Computational biology has revolutionised many fields of medicine, biology, and anatomy. The field of immunology has not been reticent to leverage mathematics to provide significant insight [98]. However, these potential benefits have not been fully realised for lymphocyte migration. Mathematical modelling is a tool that can help design and interpret assays. To receive widespread use, these tools need to be reliable, practical, and efficient. Frameworks that closely mesh with assays can place useful tools in the hands of experimentalists that routinely conduct such techniques. In a similar manner, a mathematical framework can be developed to correct for observations of scale within imaging sets and to account for differing time scales across data sets. The benefits of wide-scale adoption of scale-cognisant studies, coupled with mathematical frameworks to correct for scale phenomena, would be significant. Most importantly, the increased robustness and fidelity of cell-migration measurement would limit unintentional bias in conclusions.

In this work, we focused on the effects of scale on immune-cell migration, but scale-based phenomena in cell migration extends beyond just immune cells, and the recommendations made in this work can be applied to other migratory cell types, such as cancer cells with an invasive phenotype. This would require separate analyses to understand how scale influences the migration of the cells of interest, as well as the development of cell-specific frameworks to account for scale.

### 4.2. “Bridging the Scale Gap”: From Single-Cell to Organ-Scale Behaviour

In 2021, Fowell and Kim summarised the state of the art of effector T cell migration in the context of spatio-temporal control [6]. They commended the considerable advancements made in understanding the mechanisms of effector T cell migration but highlighted the need to bridge the divisions of scale between single-cell, high-resolution imaging and the biological complexity of whole-tissue behaviour in order to fully understand T cell localisation and positioning. This observation can be extended to investigating the whole immune system. The intricate relationship between single-cell behaviour and the behaviour of the ensemble system that drives immune response at the organ-scale is a black box. The cascades and processes that drive allergy, inflammation, and immunity emerge from a multitude of interactions between cells. These interactions affect processes, which propagate and integrate information toward responses on the organ or system scale. As a result, much of the behaviour that motivates the study of migrating immune cells is emergent in nature. Limitations with current experimental techniques restrict the ability to observe cell behaviour in spatial scales beyond several hundreds of microns. While in vitro techniques that allow the study of the migration of individual cells under closely controlled conditions exist, these emergent behaviours are traditionally only observed in vivo. The practical limits currently placed on modern microscopy make observing the scales spanned between single-cell and organ-scale behaviour experimentally unfeasible. It is apparent that novel approaches to study design are needed to push the boundaries of our understanding of complex organ-scale behaviour that drive immunological cascades and processes.

An emerging field seeks to bridge this gap and elucidate the effect of changes in individual cell migratory behaviour on homeostasis and pathology through the use a hybrid approach of experimental assays and mathematical modelling. The success of this emerging field is dependent on a collaborative effort between both experimentalists and mathematicians. Mathematical models that theoretically describe the motion of cells, with limited thought to the experimental data that can be collected, or the hypotheses that can be tested, may be of little use. Similarly, interpreting experimental data with limited consideration of the underlying models that govern cell migratory behaviour can cause critical behaviour to be missed.

As a result, an intertwined, hybrid approach of mathematical models and experiments is necessary, where one is used to inform the other to generate novel findings. As suggested by Castro et al., this requires an iterative approach of experimental observation, modelling, prediction, and experimental observation [98]. Experimental assays can suggest hypotheses that can be tested at the organ scale with mathematical models, where such models are informed by experimental data. Mathematical models can then predict the specific result of combining assays that can further elucidate specific cell dynamics and interdependent relationships with the surrounding tissue environment. An overview of three different methods of designing modelling approaches to directly integrate with existing assays are described in Figure 11, using step-based experimental data (distributions of velocity or turning angle), track-based experimental data, or continuum models (cell-density data).

Step-based data contains a list of movements from one position to the next for a collection of cells (Figure 11a). From this data, one can calculate other quantities that are useful for characterising the collective motion of cells, such as velocity, step size, and turning angle. It can be hypothesised, and those hypotheses tested, that these quantities follow probability distributions that are associated with a particular kind of motion, such as Brownian motion, Lévy walks, or correlated random walks. For example, Fricke et al. analysed the motion of T cells as they searched for DCs in the lymph node, concluding that neither Brownian motion nor Lévy walks appropriately capture the cell motion [87]. These methods have limitations, however. By lumping all observed cell motions together, it is not possible to distinguish between the heterogeneity of motion and cellular heterogeneity. For example, cells drawing from a bimodal set of velocities each step will have the same step-based velocity distribution as a bimodal distribution of cells each with fixed velocities. An assumption of cellular homogeneity is inherent in step-based methods, and this assumption should be assessed before an inference is drawn from their distributions.

Track-based data consists of time series describing the paths of cells individually (Figure 11b). A discrete, agent-based model can be used to generate an ensemble of cell tracks in silico from distributions of key parameters (informed by an underlying chemoattractant gradient), such as the waiting time between movements, step sizes, and turning angles, thereby generating a biased random walk. A prediction of mean-squared displacement versus time can be formed from the simulated cell tracks, allowing a comparison with experimental data. How biased the cell tracks are to the chemoattractant gradient can be characterised by the distribution of the aspect ratios, as in Harris et al. [86]. In the absence of chemotaxis, the distribution is expected to be symmetric about zero, whereas the presence of chemotaxis causes the distribution to shift, as illustrated in Figure 11b.

Rather than tracking individual cell positions, continuum data measure the spatial density of cells (Figure 11c). This is a mathematically convenient method to represent the locations of a collection of cells, allowing access to a range of modelling techniques such as partial differential equation models and, more specifically, to well-established continuum models of cell migration. The best known example is the Keller-Segel model [135,136], a version of which is shown in annotated form in Figure 11c. It models directed cell motion in a manner that is analogous to advection of a solute by a moving fluid, although the drift velocity of the cells is proportional to the strength of the chemoattractant gradient. Provided with an initial cell density profile at, say, t=0, a prediction of the cell density at a future time can be obtained by numerically solving the underlying differential equations repeatedly at successively later times. The final density profile can be compared with that obtained from estimating the density of cell positions from experimental data. This approach is vulnerable to the same homogeneity fallacy addressed in step-based methods. However, if work could address this weakness, this approach would mitigate some scale phenomena by reducing the required spatial resolution and allow an increase in the temporal resolution.

Recently, an example of a hybrid experimental-mathematical continuum approach can be found in the work of Hywood et al. [137]. They derived density relationships from a Fokker-Planck model, and, by estimating these relationships from microscopy data, they were able to estimate motility and chemotaxis using cell densities instead of cell tracking. They noted this approach was robust to the number of observations and agents. Approaches like these reveal how the potentially discriminating power of careful mixed-effects analysis can be directly integrated into assay design [98].

There have been demonstrations of this observation, modelling, and prediction approach since the Castro et al. review [98]. Works like Vroomans et al. can improve assay interpretation by revealing chemotaxis muting [90]. The work of Azarov et al. sought to make quantitative predictions of cognate T cell out-flux from challenged LNs using agent-based models of T cell and DC motion and interaction [138]. Other works have used in vivo measurements of T cell motion within LNs to predict the outcome of HIV infection [139]. A work recently published by this group predicted changes in T cell activation as a result of LN swelling during an antigen challenge [140]. These models bridge the gap between the temporal and spatial scales that span the micro-anatomy of cellular behaviour and the macro-scale response of organ systems, using data from one scale to provide testable hypotheses for the other. These techniques can be expanded to bridge larger gaps using in vitro cell motion to predict endpoint assays, mixing assays to maximise their strengths. Such an approach is critical for continued advancements in a field as dependent on emergent behaviour as that of immunology.

## 5. Concluding Statements

The temporal and spatial scales of an experimental assay can have a significant effect on the predicted migratory behaviour of immune cells, the influence of which is significantly underappreciated within the literature. While maximizing field of view and minimizing image interval can help mitigate scale phenomena, this is often not feasible experimentally. A scale-cognisant approach to migration assays is needed to consider the effects of scale on cell behaviour. However, studies implementing such an approach are limited and have poor agreement on how best to account for such effects. A consistent framework derived from evidence-based models is needed that can account for scale and correct observations of cell motion. This is critical to an emerging field of research that implements a hybrid approach of experimental assays and mathematical modelling. This field seeks to bridge the gap of scales between single-cell motion and organ-scale behaviour in order to improve our understanding of the complex processes that drive immune responses.

## Figures and Tables

**Figure 1 cells-10-03439-f001:**
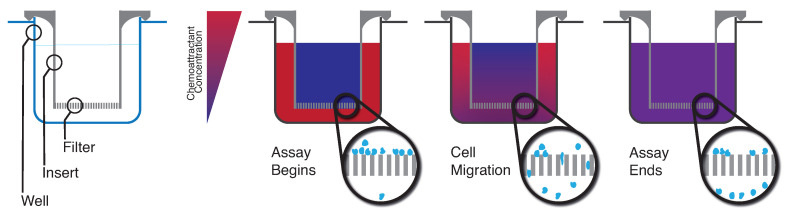
Overview schematic of a Boyden assay. Cells migrate through the filter of a cell culture insert after a chemoattractant gradient has been established. As illustrated, this chemokine gradient changes over time and is not stable long-term.

**Figure 2 cells-10-03439-f002:**
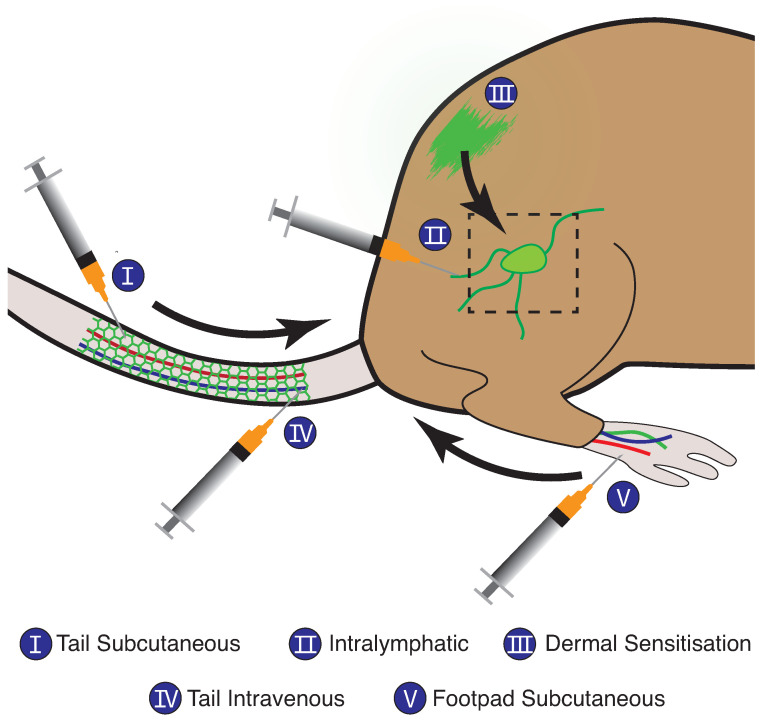
An overview of different in vivo endpoint assays, including adoptive cell transfer assays and dermal sensitisation by FITC painting. Adoptive cell transfer assays typically inject cells by three primary methods: subcutaneous injections of the tail and footpad, intravascular/intravenous injection, or intralymphatic injection. In all assays depicted, cells of interest migrate to the draining LN, which is harvested and analysed.

**Figure 3 cells-10-03439-f003:**
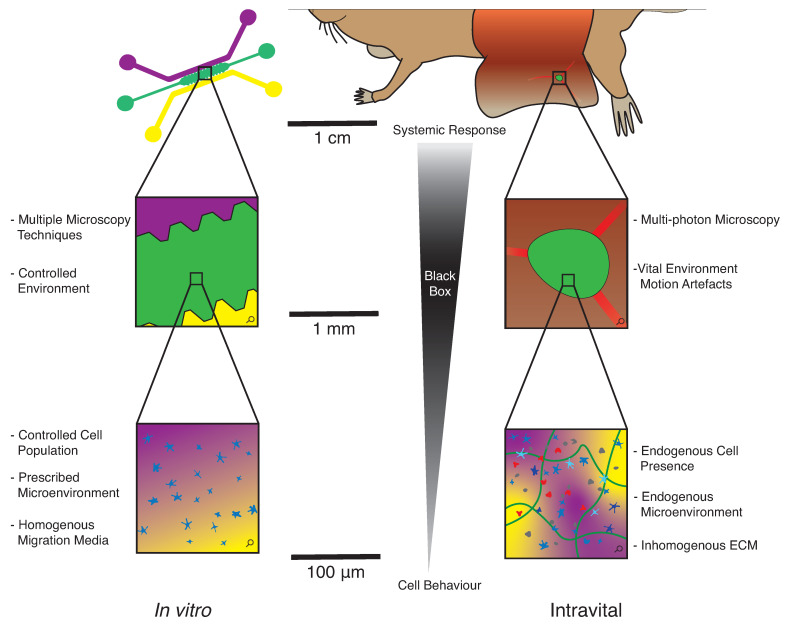
A schematic of an in vitro 3D migration assay using a microfluidic device (left) and in vivo migration assay using intravital imaging of a murine inguinal LN (right). This schematic highlights the range of scales observed and the critical features that differentiate in vivo and in vitro cell tracking. The limited observation window during live cell tracking restricts the region that can be seen to a small region some two orders of magnitude below the organism scale.

**Figure 4 cells-10-03439-f004:**
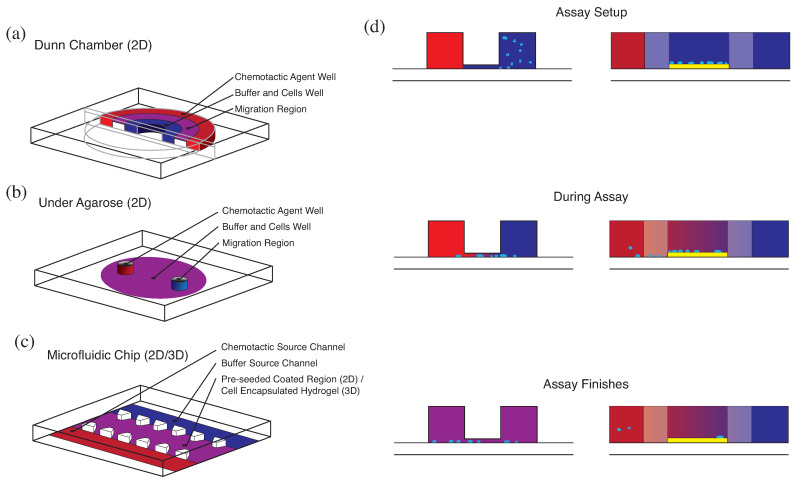
An overview of different in vitro kinetic migration assays. (**a**) The Dunn chamber consists of inner and outer annular regions that represent the buffer and chemotactic agent wells, respectively. The thin region shown in purple shows the cell migration region. (**b**) The under-agarose assay has holes punched in an agarose section poured on top of glass, where one is filled with the chemoattractant and the other with buffer. Cells migrate in the space between the agarose and the glass. (**c**) An example of a microfluidic chip, which can be used for 2D or 3D migration assays. A constant supply of buffer and chemotactic agent are maintained on either side. The central region of the chip is coated and seeded with cells (2D) or injected with a cell-encapsulated hydrogel (3D). (**d**) Three examples demonstrating cell migration and gradient formation during an unsteady or steady chemoattractant 2D migration assay. On the left is an example of unsteady assays, such as the Dunn-chamber or under-agarose assays, where the gradient changes with time. On the right is an example of a steady assay often performed using a microfluidic chip, where cells migrate along the chemoattractant gradient on the coated glass surface (shown in yellow). In these assays, a constant supply of buffer and chemoattractant maintain a stable gradient over the migration assay.

**Figure 5 cells-10-03439-f005:**
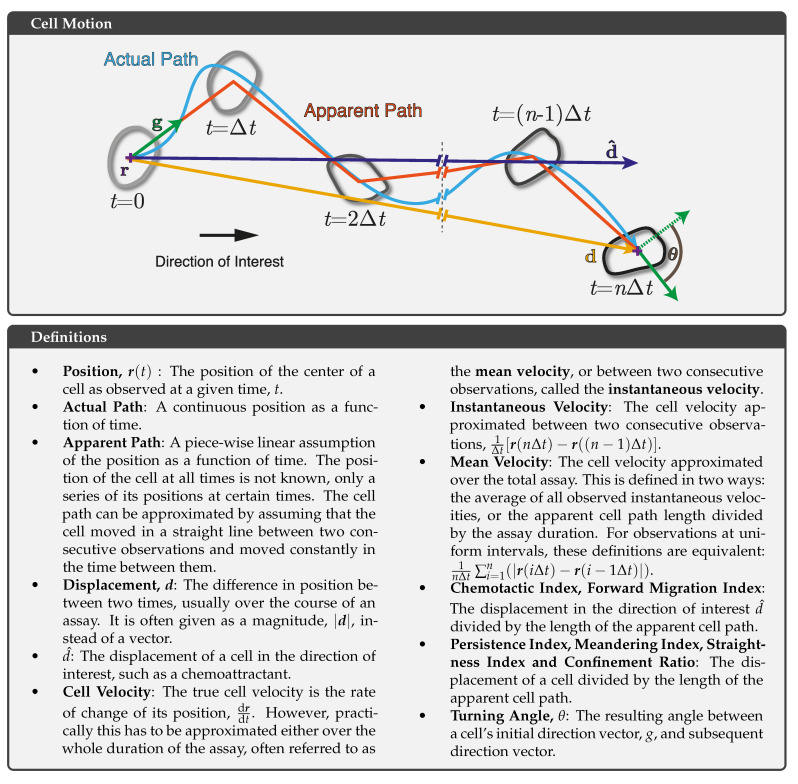
A representative schematic of a cell path, from time t=0 to t=nΔt, where Δt is the imaging interval, and *n* is the number of imaging intervals. Important cell-motility parameters commonly quantified in kinetic assays are labelled and defined.

**Figure 6 cells-10-03439-f006:**
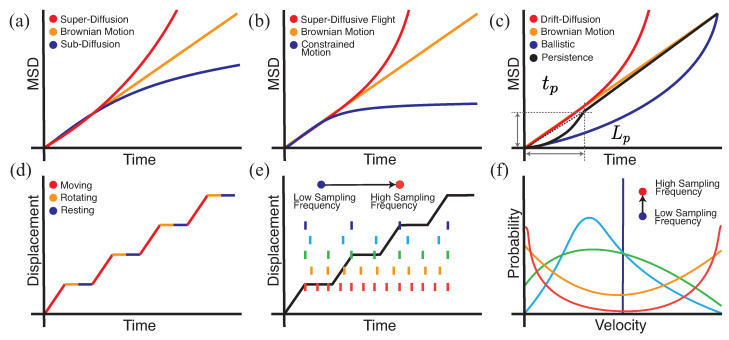
A figure showing the effect of different phenomena on the apparent displacements of cells. (**a**) The inflection of MSD plots for super–diffusive, Brownian, and sub-diffusive behaviours. (**b**) The effect of constrained motion, persistence, and super–diffusive flights on MSD. (**c**) The expected inflection of the MSD when chemotaxis is introduced both in a drift-diffusion sense and when this motion is ballistic. (**d**) The displacement–time plot for a simple multi-modal cell model, where a cell moves in a constant direction, pausing and rotating periodically. (**e**) Shows the plot from (**d**) with dashed coloured lines to indicate when an observation is taken, with frequency of observation increasing as the colour changes from blue to red. (**f**) Shows example velocity distributions reconstructed from the observation patterns shown in (**e**).

**Figure 7 cells-10-03439-f007:**
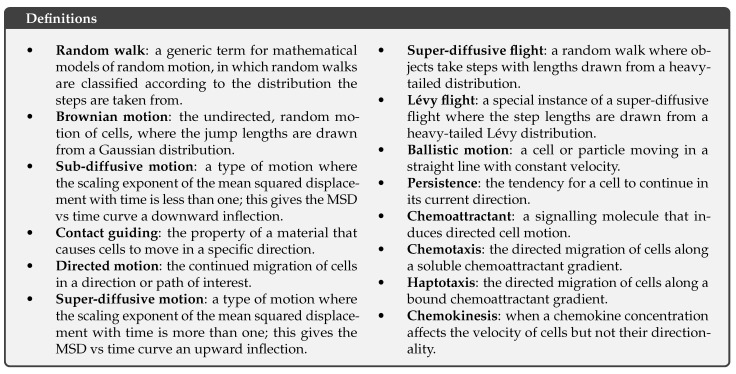
A list of common terms and their definitions used to describe cell motion.

**Figure 8 cells-10-03439-f008:**
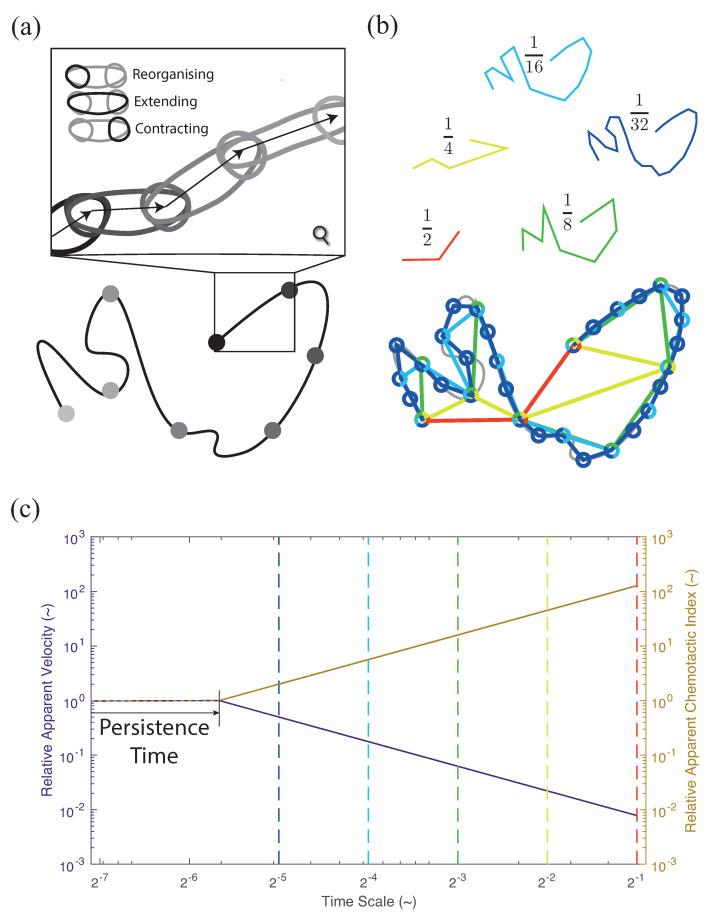
An explanatory figure demonstrating the effect of scale; (**a**) a magnified single–cell track at a temporal scale at which the mechanisms of cell migration (cell reorganising, extension, and subsequent contraction) are observable (persistence time); (**b**) the apparent path as a function of time scale for the actual cell path shown in (**a**). As the observation scale decreases, the length of the apparent path increases and its fidelity to the actual cell path is improved; (**c**) The power–law scaling of the apparent velocity and the chemotactic index as a function of time scale for the different apparent–path lengths shown in (**b**). There exists a time scale at which the cell velocity and the chemotactic index are independent of the imaging interval, defined as the persistence time.

**Figure 9 cells-10-03439-f009:**
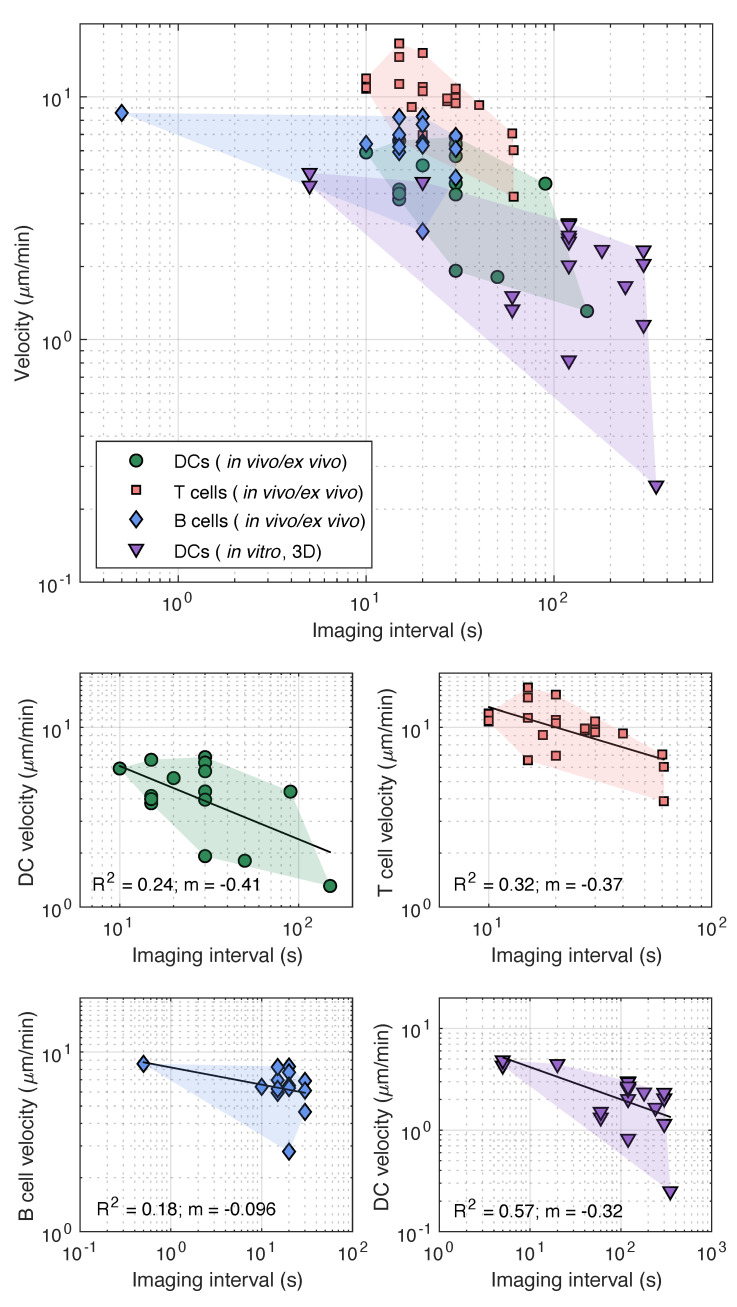
The power–law scaling of cell velocity, in μmm/min, and imaging interval, Δt, in s. For all cell types, a decreasing trend of cell velocity was observed with longer imaging intervals. Data were extracted from the references of Table 1 from both in vivo/ex vivo and in vitro assays. Velocity data of DCs, T cells, and B cells were restricted to those from wild–type mice. R–squared values are provided for each regression fit as well as the corresponding slope.

**Figure 10 cells-10-03439-f010:**
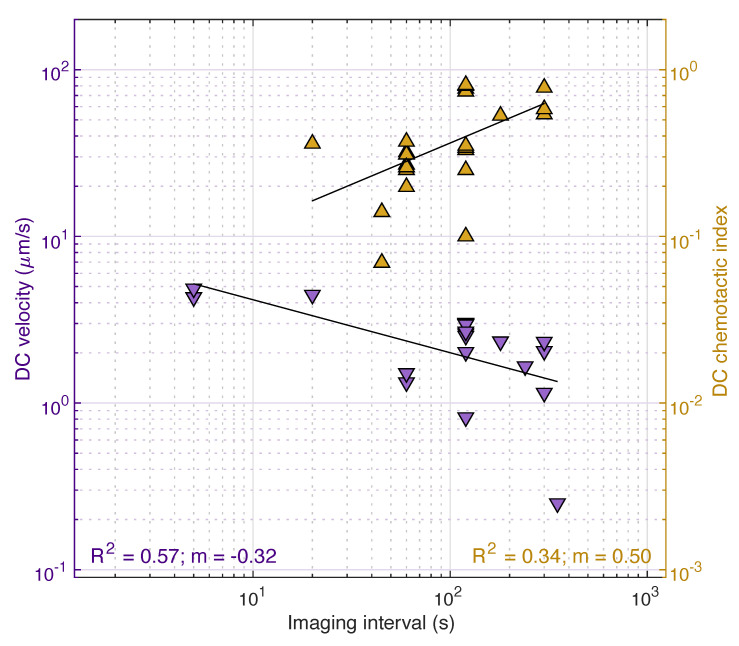
The power–law scaling of DC velocity and chemotactic index and imaging interval, Δt. An overestimation of DC chemotactic index occurs with increasing Δt, which is contrasted by an underestimation of DC velocity. Data were extracted from in vitro 3D chemotaxis assays found in the references of Table 1. The velocity and chemotactic–index data of DCs were restricted to those from wild–type mice. R–squared values are provided for each regression fit as well as the corresponding slope.

**Figure 11 cells-10-03439-f011:**
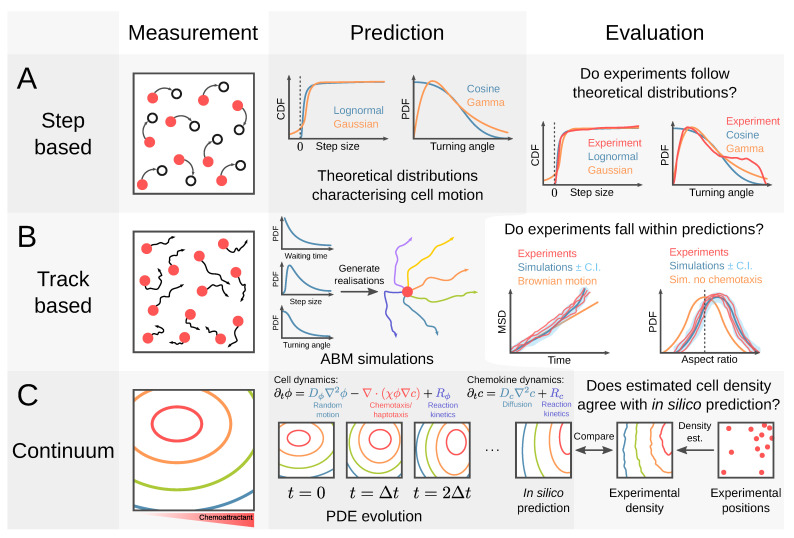
An overview of different integrated, hybrid approaches to combining experimental assays and mathematical models. Different types of experimental data that describe the motion of cells can be used including (**A**) step–based methods, (**B**) track–based methods, and (**C**) continuum methods. Mathematical models can then predict and evaluate properties of cell motion for each type of data.

**Table 1 cells-10-03439-t001:** A representative summary of varying temporal scales implemented for different in vivo, ex vivo, and in vitro kinetic assays that utilise DCs, T cells, and B cells. A wide range of imaging intervals are observed within the literature, which differs depending on the cell of interest and the selected assay type.

Type	Assay	Dimensionality	Imaging Interval	References
In vivo/ex vivo	Intravital microscopy of PLN	3D	**DCs:** 15–20 s, as high as 50 s	[32,37,41,42,43,99,100]
			**T cells:** 20 s, as low as 15 s, and as high as 60 s	[22,52,102,103,104,105]
			**B cells:** 20 s	[51,103]
	Intravital microscopy of ILN	3D	**DCs:** 30 s	[67]
			**T cells:** 30 s, as low as 10 s, and as high as 60 s	[21,106,107,108]
			**B cells:** 15–30 s, as low as 0.5 s	[20,53,109,110,111,112,133]
	Intravital microscopy of ear pinnae	3D	**DCs:** 30 s	[23,101]
			**T cells:** 60 s	[68]
	Ex vivo split-ear assay	3D	**DCs:** 15–30 s, as high as 150 s	[37,43,65]
	Excision of LN/spleen	3D	**DCs:** 30 s, as high as 90 s	[54,61,70]
			**T cells:** 10–30 s	[54,70,71,83,85,87,89,113,114]
			**B cells:** 10, 20 s	[69,85]
In vitro	Migration assay on coated plate	2D	**DCs:** 30 s	[37]
			**T cells:** 15 s	[52,71]
			**B cells:** 3 s	[115]
	Under agarose migration assay	2D	**DCs:** 60 s, as high as 300 s	[34,46,117]
			**T cells:** 60 s	[117]
	Migration assay in Dunn chambers	2D	**DCs:** 60, 600 s	[45,118]
			**T cells:** 60 s	[118]
	Migration assay in 3D matrices	3D	**DCs:** 60–120 s, as high as 300 s	[34,37,40,43,46,47,61,62,63,121,122,129,130]
			**B cells:** 60, 120 s	[20,37]
	Migration assay in microfluidic chips	2D	**DCs:** 15–30 s, as high as 180 s	[23,64,75,76,101,119]
			**T cells:** 10 s	[116]
			**B cells:** 10 s	[115]
		3D	**DCs:** 20–60 s, as low as 5 s, and as high as 300 s	[26,32,123,124,125,127,128,131,132]
			**T cells:** 30 s	[126]

## Data Availability

Not applicable.

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
