# Peer review of "The Critical Importance of Spatial and Temporal Scales in Designing and Interpreting Immune Cell Migration Assays"

_cells, 2021, doi:10.3390/cells10123439_

Round 1
Reviewer 1 Report
The manuscript presented by Dr. J. FrattoliIn and colleagues is well written and pictures are very well performed.
For reserachers who work actively in the filed of cell migration it is a good starting point and a perfect hint for experiments.
I have some suggestions:
- I'd like to suggest to the authors to add information about the importance of the cell migration of the immune cells in the immune-check points, also when they take contacts with cancer cells.
- The authors should list the strenghts and the limitations of the different approaches that they have presented.
- The authors should add information for the reader: it is important to understand if these techniques could be used also in other fields, such as for example cancer cells that have an invasive phenotype.
Author Response
The authors thank the reviewer for their suggestions, which have helped us improve the manuscript.
- I'd like to suggest to the authors to add information about the importance of the cell migration of the immune cells in the immune-check points, also when they take contacts with cancer cells.
We agree about the importance of cell migration in immune-check points and contacts with cancer cells. It would be an interesting topic to cover but would involve a repeat of the considerable effort required to summarise the findings as we did in the tables, plus Figures 8 and 9. We believe that the goal of our paper is well served by incorporating data on immune cell migration regardless of context. We hope that highlighting the issues associated with limited spatial and temporal sampling will be useful for experimental design, as well as inspiring others to re-examine cell migration data in specific contexts, such as immune checkpoints.
- The authors should list the strengths and the limitations of the different approaches that they have presented.
Considering the reviewer’s suggestion, we have strengthened our discussion of the advantages and limitations of the different approaches presented. This includes the advantages and limitations of different methods to investigate cell migration in Section 2.1, as well as when comparing various endpoint and kinetic assays. We have revised Section 3 to include a section on the phenomena of scale in endpoint assays to highlight potential scale-based limitations for this approach (lines 609-632). In the Discussion section, we have also emphasised the strengths and limitations of implementing a scale-cognisant study, highlighting the effects of phototoxicity, photobleaching, and cellular heterogeneity (lines 677-690). We have also added discussion of some potential limitations of the different mathematical models presented in Section 4.2 (lines 776-782 and lines 806-809).
- The authors should add information for the reader: it is important to understand if these techniques could be used also in other fields, such as for example cancer cells that have an invasive phenotype.
The reviewer raises an interesting discussion point regarding how a scale-cognisant framework can be applied to other cell types beyond immune cells. The reviewer is correct that the recommendations in this work can be expanded to other fields studying migratory cells. However, the development of scale-cognisant framework is highly cell-specific, including that of cancer cells with an invasive phenotype as the reviewer suggests, and will require separate analyses to understand how scale impacts such cell migration. Clarifying statements discussing these points have been added to the Discussion (lines 719-724).
Reviewer 2 Report
The manuscript entitled “Contextualisation of varying spatial and temporal scales in immune cell migration“ by Frattolin et al. reviews experimental and imaging tools to study cell migration and discusses the issues of spatial and temporal scales in migration experiments. The work is overall well-written. There are a few suggestions listed below. It is recommended that the manuscript can be published after revision.
- It is recommended that the authors can include single-cell analysis and cellular heterogeneity in migration in the review. Several relevant examples are listed below:
"Stochastic and Heterogeneous Cancer Cell Migration: Experiment and Theory"
“Cell Heterogeneity Revealed by On-Chip Angiogenic Endothelial Cell Migration”
“Functional isolation of tumor-initiating cells using microfluidic-based migration identifies phosphatidylserine decarboxylase as a key regulator”
“Heterogeneous T cell motility behaviors emerge from a coupling between speed and turning in vivo”
- For the imaging interval, it is recommended that the authors can discuss the issues of photobleaching and phototoxicity, which might be detrimental in migration experiments.
Author Response
The authors thank the reviewer for their suggestions, which have helped us improve the manuscript.
- It is recommended that the authors can include single-cell analysis and cellular heterogeneity in migration in the review.
The reviewer raises an important point regarding the effects of single-cell analysis in assessing cell migration. As a result of the reviewer’s recommendation, we have added to Section 2.1 how endpoint assays are limited in their ability to distinguish cellular heterogeneity, whereas kinetic assays can incorporate both single-cell analysis and cellular heterogeneity (lines 98-102, 188-190). In the Discussion, we have also expanded upon how cellular heterogeneity can influence the selection of an appropriating imaging interval (lines 670-676). In addition, we have edited Section 4.2 to highlight that continuum-based and step-based models of cell motion, including those proposed by Hywood et al., are unable to capture cellular heterogeneity, which requires the use of track-based methods instead (lines 776-782, 806-809).
- For the imaging interval, it is recommended that the authors can discuss the issues of photobleaching and phototoxicity, which might be detrimental in migration experiments.
Photobleaching and phototoxicity are indeed important factors to consider when selecting an appropriate imaging interval for cell migration experiments. We have added a discussion of the issues of photobleaching and phototoxicity within Section 4.1 (lines 677-690).
Round 2
Reviewer 2 Report
The authors addressed the comments from the reviewers.
Author Response
The authors thank the reviewer for their acknowledgement of the revisions.